# Assembly rules for GABA$_A$ receptor complexes in the brain

**James S Martenson[1,2], Tokiwa Yamasaki[1,2], Nashid H Chaudhury[1,2], David Albrecht[1,2], Susumu Tomita[1,2]***

[1]Department of Cellular and Molecular Physiology, Yale University School of Medicine, New Haven, United States; [2]Department of Neuroscience, Program in Cellular Neuroscience, Neurodegeneration and Repair, Interdepartmental Neuroscience Program, Yale University School of Medicine, New Haven, United States

**Abstract** GABA$_A$ receptor (GABA$_A$R) pentamers are assembled from a pool of 19 subunits, and variety in subunit combinations diversifies GABA$_A$R functions to tune brain activity. Pentamers with distinct subunit compositions localize differentially at synaptic and non-synaptic sites to mediate phasic and tonic inhibition, respectively. Despite multitudes of theoretical permutations, limited subunit combinations have been identified in the brain. Currently, no molecular model exists for combinatorial GABA$_A$R assembly in vivo. Here, we reveal assembly rules of native GABA$_A$R complexes that explain GABA$_A$R subunit subcellular distributions using mice and *Xenopus laevis* oocytes. First, $\alpha$ subunits possess intrinsic signals to segregate into distinct pentamers. Second, $\gamma$2 is essential for GABA$_A$R assembly with Neuroligin-2 (NL2) and GARLHs, which localize GABA$_A$Rs at synapses. Third, $\delta$ suppresses $\alpha$6 synaptic localization by preventing assembly with GARLHs/NL2. These findings establish the first molecular model for combinatorial GABA$_A$R assembly in vivo and reveal an assembly pathway regulating GABA$_A$R synaptic localization.

*For correspondence:
susumu.tomita@yale.edu

Competing interests: The authors declare that no competing interests exist.

## Introduction

Heteromeric ion channels are tailored from subunit arrays to ensure precision in channel function and exquisite control over membrane potential. In the brain, fast inhibition of synaptic membrane depolarization is mediated principally by the binding of GABA to ionotropic GABA receptors (GABA$_A$Rs), hetero- or homo-pentamers consisting of combinations of six $\alpha$, three $\beta$ and ten non-$\alpha$/$\beta$ subunits (*Barnard et al., 1998*; *Olsen and Sieghart, 2008*). While a huge number of permutations are theoretically possible, only a fraction are observed in neural tissues, with just a handful of major GABA$_A$R subtypes dominating (*Barnard et al., 1998*; *McKernan and Whiting, 1996*; *Olsen and Sieghart, 2008*). This preferential subunit assembly results in GABA$_A$Rs with specialized localization and function. For example, in cerebellar granule cells, $\alpha$1/$\beta$/$\gamma$2-containing GABA$_A$Rs localize at synapses and mediate phasic inhibition, whereas $\alpha$6/$\beta$/$\delta$-containing GABA$_A$Rs localize at extrasynaptic sites and mediate tonic inhibition (*Günther et al., 1995*; *Jones et al., 1997*; *Mihalek et al., 1999*; *Nusser et al., 1998*). Beyond these cardinal cases, there are numerous long-standing examples of particular GABA$_A$R subtypes whose subunit compositions, distributions and functions have been described (*Fritschy et al., 2012*; *Olsen and Sieghart, 2008*; *Sigel and Steinmann, 2012*). For example, the major GABA$_A$R subtypes contain at most one non-$\alpha$/$\beta$ subunit, making non-$\alpha$/$\beta$ subunits mutually exclusive within a pentamer (*Araujo et al., 1998*; *Jechlinger et al., 1998*). By contrast, it remains unclear how the majority of GABA$_A$R pentamers incorporate two $\alpha$ subunits of a single isoform (*Barnard et al., 1998*), or which non-$\alpha$/$\beta$ subunit dictates pentamer assembly of each $\alpha$ and $\beta$ isoform in vivo. Thus, the rules constraining GABA$_A$R assembly, and the precise mechanism by which GABA$_A$R subtype determines distribution, have not been fully revealed.

Ion channels often function with auxiliary subunits that modulate localization and/or channel properties (*Jackson and Nicoll, 2011*; *Yan and Tomita, 2012*). AMPA receptors form a complex with TARP auxiliary subunits, which are required for AMPA receptor synaptic localization. Similarly, GARLH putative auxiliary subunits of GABA$_A$Rs were recently identified in the brain (*Yamasaki et al., 2017*). GARLHs form complexes with GABA$_A$Rs and the inhibitory synaptic cell adhesion molecule Neuroligin-2 (NL2), and are essential for synaptic localization and inhibitory postsynaptic currents (IPSCs), but not GABA$_A$R activity at the cell surface in primary hippocampal neurons and the hippocampus. Furthermore, synaptic localization of the inhibitory scaffolding molecule gephyrin requires GARLH expression in hippocampus (*Yamasaki et al., 2017*). Thus, GARLHs play a major role in the synaptic localization and downstream signaling of GABA$_A$Rs. However, the subunit specificity of GABA$_A$R assembly with GARLH/NL2 in vivo is not fully understood.

Here, we aimed to uncover the rules determining which subunits coassemble within a single complex, and which segregate into distinct complexes. To address this question, we examined the formation of GABA$_A$Rs and their association with GARLH/NL2 in heterologous cells and in vivo using various knock out mice. Our results reveal three novel assembly rules for GABA$_A$Rs and GARLH/NL2. First, α1 and α6 subunits possess intrinsic signals to preferentially segregate into distinct pentamers. Second, γ2 is required for native GABA$_A$Rs to assemble with GARLH/NL2. Third, δ inhibits assembly of α6 with γ2 and thus GARLH/NL2. These findings establish a simple model for restricted combinations of subunits in GABA$_A$R pentamers in vivo and reveal an assembly pathway that increases GABA$_A$R synaptic targeting and synaptic transmission in the absence of δ.

## Results

### Distinct compositions of GARLHed and GARLHless GABA$_A$Rs

As an in vivo model for GABA$_A$R assembly, we focused on cerebellar granule cells, which predominantly express two distinct GABA$_A$R subtypes: α1/β/γ2- and α6/β/δ-containing GABA$_A$Rs (*Jechlinger et al., 1998*; *Nusser et al., 1999*). We analyzed constituents of native GABA$_A$R complexes using blue native PAGE (BN-PAGE). BN-PAGE preserves protein complexes but cannot accurately measure their molecular weights, because in contrast to SDS-PAGE, protein complex structure affects migration on BN-PAGE (*Kim et al., 2010*; *Schägger et al., 1994*). For example, AMPA receptors lacking their N-terminal domains migrate at 55 kDa on SDS-PAGE, while a tetramer of these subunits migrates at 480 kDa on BN-PAGE, roughly twice the expected 220 kDa for a tetramer of 55 kDa subunits (*Kim et al., 2010*).

We solubilized mouse cerebellar membranes with lauryl maltose-neopentyl glycol (MNG), followed by BN-PAGE and western blotting. We found that all δ and most α6 migrated at 480 kDa, whereas nearly all γ2 and most α1 migrated at 720 kDa (*Figure 1*). By contrast, β2/3 migrated equally at 480 and 720 kDa (*Figure 1*). In the brain, GABA$_A$Rs assemble with GARLHs and NL2 to form a tripartite complex that migrates at 720 kDa on BN-PAGE (*Yamasaki et al., 2017*). Consistently, we found that both GARLH4 and NL2 also co-migrated at 720 kDa. Thus, endogenous GABA$_A$R subunits segregate into two major complexes—a GARLH/NL2-associated (GARLHed) α1/β/γ2-containing complex migrating at 720 kDa, and a GARLHless α6/β/δ-containing GABA$_A$R migrating at 480 kDa (*Figure 1*).

### α1 and α6 subunits possess intrinsic signals to preferentially segregate into distinct pentamers

To reveal rules for GABA$_A$R assembly, we turned to cRNA-injected *Xenopus laevis* oocytes as a heterologous expression system. We first confirmed assembly in this system of the GABA$_A$R subunits α1, β2 and HA-tagged γ2 (HAγ2, in which the HA epitope was inserted after the γ2 signal sequence) by analyzing Triton X-100-solubilized oocyte membranes using BN-PAGE. We observed α1/β2 and α1/β2/HAγ2 hetero-oligomers at 520 kDa, slightly higher than the 480 kDa complex in the brain (*Figure 2A*). This size difference corresponds with differences in the molecular weights of the GABA$_A$R subunits expressed in oocytes and in the brain on SDS-PAGE and may be caused by differences in species, alternative splicing or post-translational modification (*Yamasaki et al., 2017*). Corresponding to the 520 kDa complexes, we observed α1/β2- and α1/β2/γ2-mediated GABA-evoked currents (*Figure 2—figure supplement 1A*). In addition, we detected weakly expressed β2/HAγ2

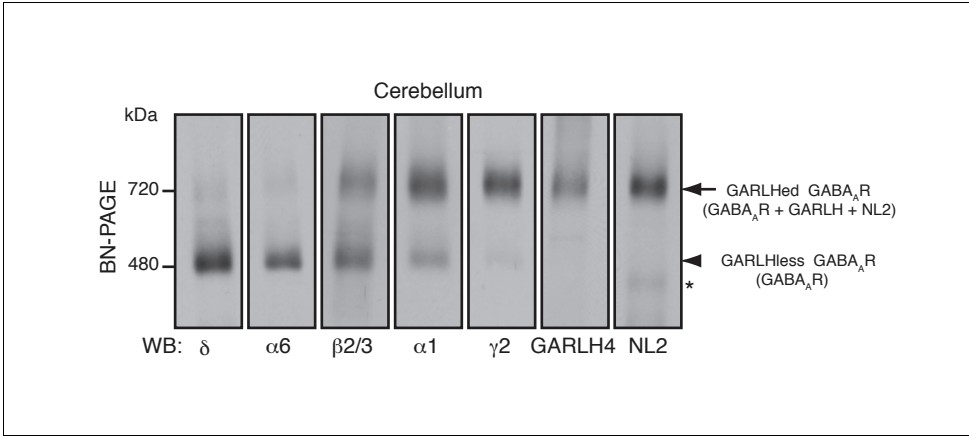

**Figure 1.** Distinct compositions of GARLHed and GARLHless GABA$_A$Rs. Cerebellar membranes solubilized with lauryl maltose-neopentyl glycol (MNG) were subjected to BN-PAGE. The α6 and δ GABA$_A$R subunits preferentially migrated at 480 kDa, while γ2 and α1 as well as GARLH4 and neuroligin-2 (NL2) predominantly migrated at 720 kDa. β2/3 signal was observed equally at 480 and 720 kDa. The arrow and arrowhead indicate the GARLHed and GARLHless GABA$_A$R, respectively, while the asterisk (*) denotes the NL2 band without GABA$_A$Rs. The images are representative of three independent experiments.

hetero-oligomers at 600 kDa (*Figure 2A*) and tiny β2/γ2-mediated GABA-evoked currents (*Figure 2—figure supplement 1A*), whereas neither α1, β2 nor HAγ2 homomers were detected (*Figure 2A* and *Figure 2—figure supplement 1A*). These results demonstrate the assembly of functional GABA$_A$R complexes in cRNA-injected oocytes.

To reveal the number of subunits comprising the 520 kDa complex in cRNA-injected oocytes, we compared the migration of an α1/HAβ2/γ2 hetero-oligomer and a pentameric GABA$_A$R concatemer, HAβ2-α1-HAβ2-α1-γ2 (HA$_2$Pent), that was previously shown to be functional (*Baur et al., 2006*). On SDS-PAGE, both monomeric HAβ2 and HA$_2$Pent were detected at their expected molecular weights of 50 kDa and 260 kDa, respectively (*Figure 2B*). On BN-PAGE, HA$_2$Pent migrated at 520 kDa, similar to α1/HAβ2/γ2 GABA$_A$Rs, although the signal was weak, likely due to a difference in pentamer expression levels and HA epitope accessibility (*Figure 2B*). An anti-α1 N-terminus antibody that recognizes monomeric but not concatenated α1 detected α1/HAβ2/γ2, but not HA$_2$Pent, at 520 kDa (*Figure 2B*), confirming the absence of monomeric α1 in oocytes expressing the concatenated pentamer. We also examined a concatenated GABA$_A$R trimer, HAβ2-α1-β2, which migrated at 520 kDa only when co-expressed with both α1 and γ2 monomers (*Figure 2—figure supplement 1B*). HAβ2-α1-β2 alone was only detectable following long exposures and migrated at 400 and 600 kDa, presumably corresponding to the trimer and a dimer of trimers (hexamer), respectively (*Figure 2—figure supplement 1B*). Thus, we concluded that the 520 kDa complex in cRNA-injected oocytes consists of a GABA$_A$R pentamer.

The majority of GABA$_A$R pentamers in vivo incorporate two α subunits of a single isoform (*Barnard et al., 1998*). Consistent with this, we found that α1 and α6 preferentially incorporate into GARLHed and GARLHless GABA$_A$R complexes, respectively, and thus are largely segregated in vivo (*Figure 1*). However, it is unclear what rule determines α1 and α6 segregation. Because γ2 and δ are mutually exclusive (*Araujo et al., 1998*; *Jechlinger et al., 1998*), one possibility is that preferential assembly of α1 with γ2 and α6 with δ ensures α1/α6 segregation. Alternatively, α1 and α6 may segregate independent of non-α/β subunits. To directly test this, we analyzed assembly of both α isoforms with β2 in the absence of non-α/β subunits using an antibody shift assay. An antibody shift assay is a powerful assay to confirm the existence of a protein in a complex on BN-PAGE (*Figure 2—figure supplement 1C*). In this method, we pre-incubate lysate with an antibody prior BN-PAGE and western blotting. Antibody-bound complexes will migrate at a higher molecular weight on the BN-PAGE gel, indicating the existence of the antigen in the protein complex (*Figure 2—figure supplement 1C*). It is critical that the antibody for pre-incubation and the antibody for western blotting come from different species, because pre-incubated antibodies can also be detected by the secondary antibody during western blot analysis. We expressed both α isoforms (α1 and HA-tagged

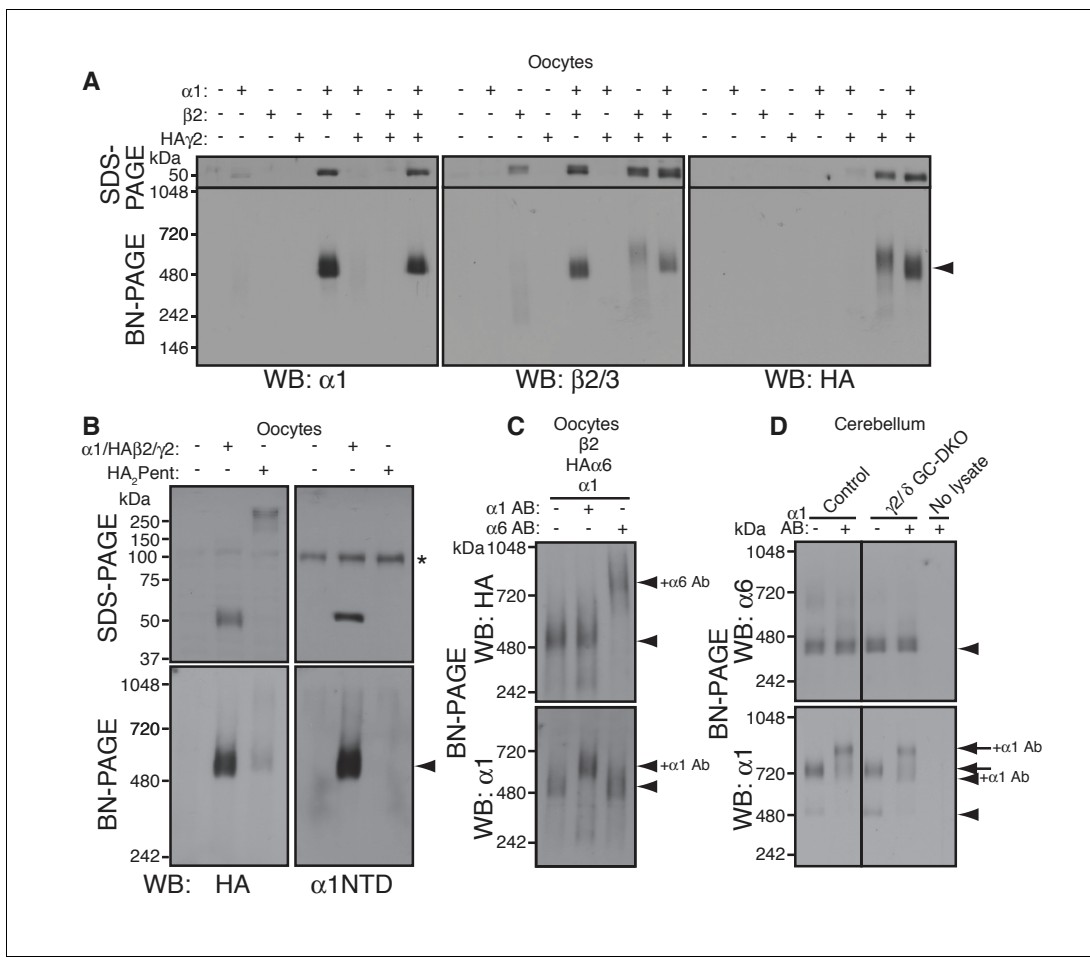

**Figure 2.** α1 and α6 subunits segregate into distinct pentamers independent of non-α/β subunits. (**A**) Reconstitution of GABA_AR assembly in *Xenopus laevis* oocytes. Membranes from oocytes injected with cRNAs encoding the indicated subunits were solubilized with Triton X-100 and subjected to SDS- and BN-PAGE. The GABA_AR at 520 kDa was reconstituted by co-expression of α1 and β2 or α1, β2 and HA-tagged γ2 in oocytes injected with the corresponding cRNAs (0.55 ng ea). Co-expressing β2 and HAγ2 produced a weak band at 600 kDa. The images are representative of two independent experiments. (**B**) Co-migration of a GABA_AR oligomer and concatenated pentamer. Membranes from cRNA-injected oocytes were solubilized with Triton X-100 and subjected to SDS- and BN-PAGE. An α1/HAβ2/γ2 GABA_AR oligomer and a concatenated pentamer, HAβ2-α1-HAβ2-α1-γ2 (HA_2Pent), migrated at 520 kDa. Monomers and HA_2Pent were visualized at the expected molecular weights on SDS-PAGE. An anti-α1 antibody that recognizes the N-terminus of mature α1 proteins (α1NTD) detects the monomeric, but not concatenated, α1 subunit. The asterisk (*) denotes a nonspecific band observed on all lanes, indicating that the band is not heterologously expressed GABA_AR subunit. The images are representative of three independent experiments. (**C**) GABA_AR complexes from oocytes co-injected with cRNAs of HAα6, α1 and β2 were examined by antibody shift assay. An anti-α6 antibody shifted up HAα6 but not α1 signal, whereas an anti-α1 antibody shifted up α1 but not HAα6 signal. The images are representative of three independent experiments. (**D**) GABA_AR complexes in cerebella from control and γ2/δ GC-DKO mice were examined by antibody shift assay on BN-PAGE. Addition of anti-α1 antibody shifted up α1 signal at 480 and 720 kDa in both genotypes. In contrast, in both genotypes, α6 signal was not shifted by addition of anti-α1 antibody. The images are representative of three independent experiments. The arrow and arrowhead indicate the GARLHed and GARLHless GABA_AR, respectively, and antibody bound complexes are indicated.

The online version of this article includes the following figure supplement(s) for figure 2:

**Figure supplement 1.** GABA_AR assembly in cRNA-injected oocytes and characterization of knockout mice.

α6, permitting use of rabbit anti-α6 and mouse anti-HA antibodies for HAα6 shift and detection, respectively) and β2 subunits. α1 and HAα6 ran as 520 kDa pentamers on BN-PAGE (*Figure 2C*, first lane). Addition of anti-α6 antibody shifted up only the HAα6 signal, but not the α1 signal (*Figure 2C*, third lane). Conversely, addition of anti-α1 antibody shifted up only the α1 signal, but not the HAα6 signal (*Figure 2C*, second lane). The results indicate that α1 and HAα6 segregate independent of non-α/β subunits when co-expressed with β2, and thus their segregation is encoded by the α subunits themselves.

To test if this was also true in vivo, we used the antibody shift assay to analyze α1/α6 segregation in cerebellum lacking both the γ2 and δ subunits. Because conventional γ2 knockout (KO) mice show postnatal lethality (*Günther et al., 1995*), we crossed double conditional *Gabrg2*<sup>fl/fl</sup>/*Gabrd*<sup>fl/fl</sup> mice with transgenic mice expressing Cre recombinase under the *Gabra6* promoter (*Fünfschilling and Reichardt, 2002*; *Lee and Maguire, 2013*; *Schweizer et al., 2003*) (*Figure 2—figure supplement 1D*) (see Materials and methods), resulting in viable γ2/δ granule cell (GC)-specific double knock out (DKO) mice that displayed no changes in body weight (*Figure 2—figure supplement 1E*). In both control and γ2/δ GC-DKO cerebella, addition of an anti-α1 antibody did not supershift the α6 signal, but did supershift the α1 signal expressing mostly in the Purkinje cells, suggesting that α1 does not incorporate into α6-containing complexes even in the absence of γ2 and δ (*Figure 2D*). Thus, in vivo, the segregation of α1 and α6 into distinct GABA<sub>A</sub>R complexes is independent of γ2 and δ.

## γ2 is essential for assembly of the native GARLHed GABA<sub>A</sub>R complex

We next explored the preferential association of γ2 subunits with GARLH/NL2. Although we have previously shown that γ2 promotes GARLH4/NL2 assembly with α1/β2/γ2-containing GABA<sub>A</sub>Rs in heterologous systems (*Yamasaki et al., 2017*), whether γ2 is necessary for assembly of native GARLHed complexes in neurons remains unclear. If γ2 is necessary for GABA<sub>A</sub>R assembly with GARLH4, it should be present in all GARLHed complexes. We first asked to what extent α1 and β2 subunits assemble with γ2 in oocytes using an antibody shift assay. When α1 and β2 were coexpressed without γ2, they formed a 520 kDa GABA<sub>A</sub>R that was not affected by pre-incubation with an anti-γ2 antibody (*Figure 3A*). By contrast, when γ2 was coexpressed with α1 and β2, the three subunits formed a 520 kDa GABA<sub>A</sub>R that was completely supershifted by the anti-γ2 antibody (*Figure 3A*), indicating that all α1 and β2 assemble with γ2 in our oocyte system.

To determine what portion of GARLHed GABA<sub>A</sub>Rs contain γ2 in the cerebellum, we performed an antibody shift assay of cerebellar lysate using BN-PAGE, and blotted for β2/3, which is present in both GARLHed complexes and GARLHless GABA<sub>A</sub>Rs in the cerebellum (*Figure 1*). Addition of only an anti-γ2 antibody supershifted most or all the GARLHed complex, and only a small fraction of the GARLHless GABA<sub>A</sub>R (*Figure 3B*). On the other hand, addition of an anti-δ antibody specifically supershifted most of the GARLHless GABA<sub>A</sub>R (*Figure 3B*). Addition of both anti-γ2 and anti-δ antibodies supershifted both the GARLHed complex and GARLHless GABA<sub>A</sub>R (*Figure 3B*). A schematic diagram of this result is provided (*Figure 3—figure supplement 1*). These results suggest that most of the GARLHed complexes and GARLHless GABA<sub>A</sub>Rs in the cerebellum contain γ2 and δ, respectively.

To assess if γ2 is necessary for GABA<sub>A</sub>R assembly with GARLH/NL2 in neurons, we examined assembly specifically in γ2 deficient cerebellar granule cells, since elimination of γ2 from all cells causes mouse lethality (*Günther et al., 1995*). We cultured granule cells from conditional *Gabrg2*<sup>fl/fl</sup> mice expressing tamoxifen-inducible Cre recombinase (CreERT) under the CAG promoter (*Hayashi and McMahon, 2002*), as well as from control *Gabrg2*<sup>fl/fl</sup> littermates not expressing CreERT, and treated both with 4-hydroxytamoxifen (4-OHT) from DIV 1.5–3. In control primary cultures, both α1 and γ2 incorporated equally into GARLHed complexes and GARLHless GABA<sub>A</sub>Rs (*Figure 3C*). On the other hand, in *Gabrg2*<sup>fl/fl</sup> cultures expressing CreERT, both γ2 expression and GARLHed complexes were eliminated (*Figure 3C*). Combining our new finding that γ2 is required in vivo for assembling the native GABA<sub>A</sub>R complex (*Figure 3C*) with the finding from *Yamasaki et al. (2017)* that γ2 is required for reconstituting the native GABA<sub>A</sub>R complex in a heterologous system, we conclude that the association of native GABA<sub>A</sub>Rs with GARLHs requires γ2.

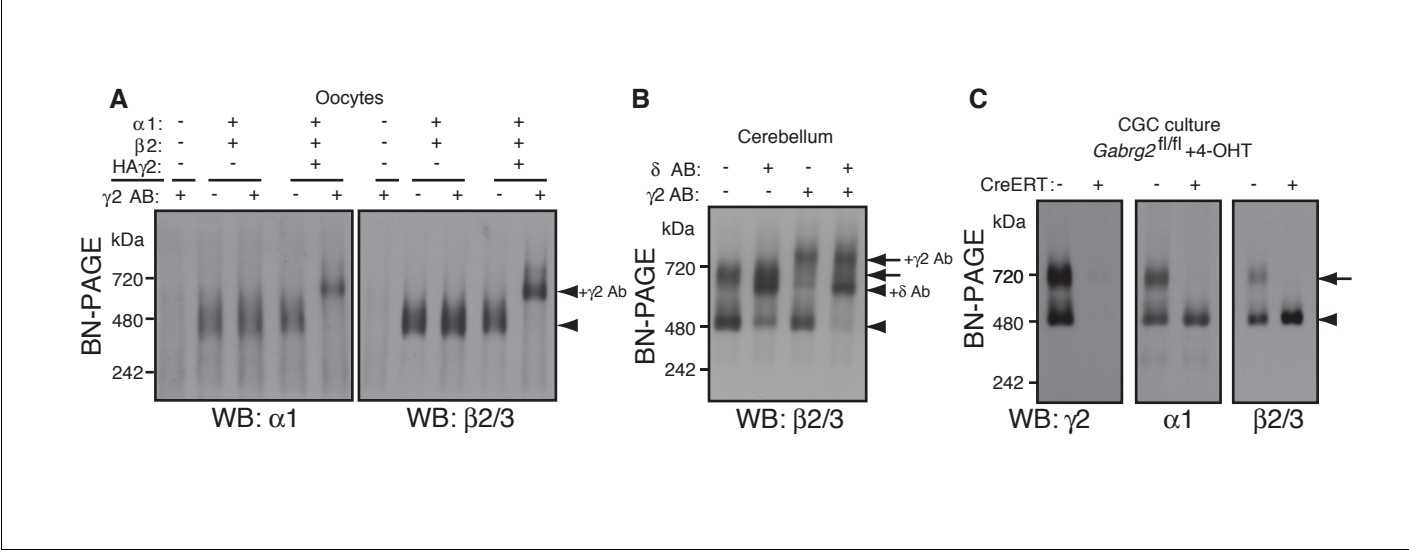

**Figure 3.** γ2 is required for assembly of the native GARLHed complex. (**A**) Membranes from cRNA-injected oocytes (0.18 ng ea) were solubilized in Triton X-100 and analyzed by BN-PAGE. α1 and β2 with or without HA-tagged γ2 migrated at 480 kDa. Addition of an anti-γ2 antibody induced an upward shift of GABA$_A$Rs from α1/β2/HAγ2-injected oocytes but had no effect on GABA$_A$Rs from α1/β2-injected oocytes, indicating nearly complete incorporation of HAγ2 into GABA$_A$R pentamers. The images are representative of two independent experiments. (**B**) GABA$_A$R complexes in cerebellum were examined by antibody shift assay. In cerebellum, anti-δ antibody caused most β2/3 signal at 480 kDa to shift up, but did not affect β2/3 signal at 720 kDa. In contrast, anti-γ2 antibody shifted up β2/3 signal at 720 kDa. When anti-δ and anti-γ2 antibodies were combined, both 480 and 720 kDa bands shifted up almost completely. The images are representative of two independent experiments. (**C**) Primary cultured cerebellar granule cells were prepared from conditional γ2 knockout mice with or without a transgene encoding tamoxifen-inducible Cre recombinase (CreERT), and treated with 4-hydroxytamoxifen (4-OHT) from DIV1.5 to DIV3. At DIV9, cell membranes were solubilized in MNG and examined by BN-PAGE. In neurons expressing CreERT, γ2 was eliminated, and α1 and β2 at 720 kDa collapsed to 480 kDa. The images are representative of three independent experiments. The arrow and arrowhead indicate the GARLHed and GARLHless GABA$_A$R, respectively, and antibody-bound complexes are indicated.

The online version of this article includes the following figure supplement(s) for figure 3:

**Figure supplement 1.** A schematic diagram of an antibody shift assay for distinct GABA$_A$R complexes on BN-PAGE.

## γ2 is essential for GABA$_A$R synaptic localization in the adult brain

γ2 is required for GABA$_A$R synaptic localization in cultured cortical neurons (*Essrich et al., 1998*) and neonatal dorsal root ganglion neurons (*Günther et al., 1995*). However, the role of γ2 in GABA$_A$R synaptic localization in the adult brain remains unclear. To examine GABA$_A$R synaptic localization in the adult brain in the absence of γ2, we turned to cerebellar granule cell (GC)-specific conditional γ2 knockout (KO) mice (γ2 GC-KO) obtained by crossing conditional *Gabrg2*$^{fl/fl}$ mice with *Gabra6* promoter-Cre transgenic mice (*Fünfschilling and Reichardt, 2002*; *Schweizer et al., 2003*). These mice are viable with no change in body weight (*Figure 2—figure supplement 1E*). We previously showed that in γ2 GC-KO mice, the protein levels of GARLH4 and NL2 are reduced in total cerebellar lysate, while the protein levels of GARLH4, NL2 and α1 are reduced in the glomerular postsynapse-enriched fraction (*Yamasaki et al., 2017*).

To assess the role of γ2 in GABA$_A$R synaptic localization in the adult brain directly, we examined the distribution of GABA$_A$Rs in γ2 GC-KO granule cells in vivo using immunohistochemistry. We confirmed loss of γ2 expression specifically in the granular layer of adult γ2 GC-KO mice (*Figure 4A*). By contrast, overall intensity of α1 and β2 signal was not noticeably altered (*Figure 4A*). High-magnification images revealed the doughnut-like structure of cerebellar glomeruli (*Figure 4B*). A central hole corresponds to an excitatory input and is surrounded by excitatory synapses on the glomerular interior, while inhibitory inputs form synapses on the glomerular periphery (*Jakab and Hámori, 1988*). In control mice, α1 formed clusters apposed to inhibitory presynaptic VGAT on the glomerular periphery, and also displayed a weaker, diffuse distribution across the entire glomeruli that overlapped with a glomerular marker, the NMDA receptor subunit GluN1 (*Figure 4B and C*). By contrast, in the γ2 GC-KO mice, the fraction of α1 colocalized with VGAT was substantially reduced, while the fraction of GluN1 colocalized with α1 was substantially increased (*Figure 4B and C*). We

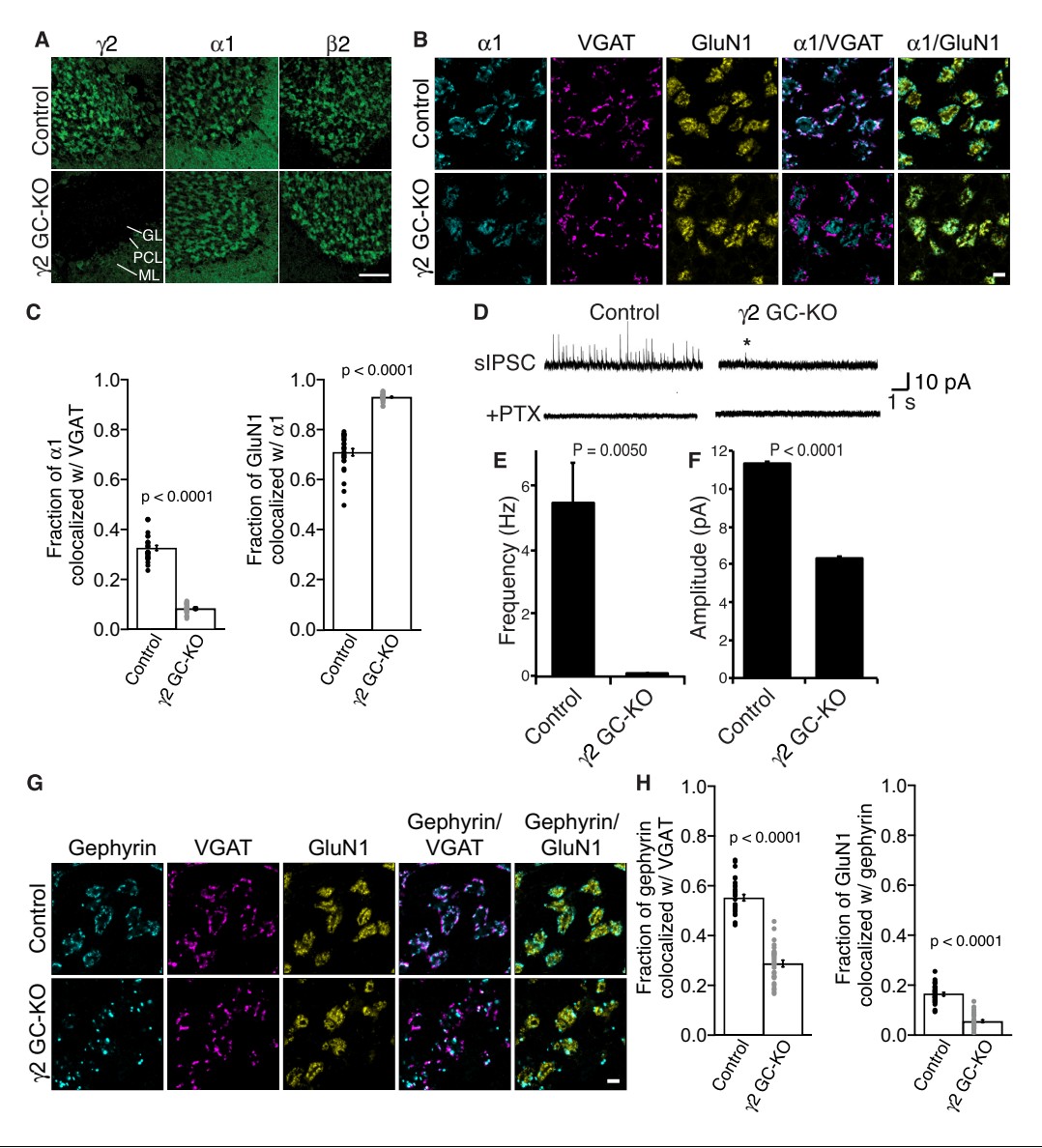

**Figure 4.** γ2 is essential for GABA$_A$R synaptic localization in the brain. (**A, B**) Localization of GABA$_A$R subunits in the cerebellar granule cell (GC)-γ2 knockout (KO) mice and age matched controls without Cre expression (Control). Inhibitory presynaptic VGAT and excitatory postsynaptic GluN1 were co-stained. (**A**) Loss of γ2 was observed specifically in the granular layer in γ2 GC-KO mice, whereas α1 and β2 remained. The images are representative of four independent experiments. (**B and C**) High-magnification representative images showed protein distribution on each glomerulus. Inhibitory inputs project to outer edges of the glomerulus, whereas excitatory inputs project to inner edges of the glomerulus. In the γ2 GC-KO, the fraction of α1 colocalized with VGAT was reduced, whereas the fraction of GluN1 colocalized with α1 was increased (n = 30 areas/2 animal each). (**D–F**) Spontaneous inhibitory postsynaptic currents (sIPSCs) were recorded from granule cells in acute cerebellar slices, and representative traces are shown (**D**). In γ2 GC-KO mice, sIPSC frequency (**E**) and amplitude (**F**) were dramatically reduced, but not completely eliminated (n = 4 bins (**E**), n = 69–1740 events (**F**), see Materials and methods). The asterisk indicates a sIPSC recorded from a γ2 GC-KO mouse. Picrotoxin (100 μM) blocked all sIPSCs. (**G and H**) Representative images show localization of gephyrin in γ2 GC-KO and control mice. Gephyrin colocalized with VGAT at the glomerular periphery in controls. In the γ2 GC-KO, the fraction of gephyrin colocalized with VGAT was reduced, and at the same time, the fraction of GluN1 colocalized with gephyrin was reduced (n = 30 areas/2 animal each). Scale bars: 60 μm (**A**), 5 μm (**B, G**). Data are given as mean ± s.e.m.; p values were determined using student's t test.

next evaluated GABA$_A$R-mediated synaptic transmission. Because the frequency of miniature IPSCs (mIPSCs) was low in granule cells from acute cerebellar slices, we measured GABA$_A$R-mediated spontaneous IPSCs (sIPSCs). GABA$_A$R-mediated sIPSCs were almost completely eliminated in γ2 GC-KO mice (*Figure 4D and E*). The rare residual sIPSCs in γ2 GC-KO mice displayed decreased amplitude (*Figure 4F*). Picrotoxin eliminated sIPSCs (*Figure 4D*). The results indicate that the vast majority of GABA$_A$R-mediated synaptic events require γ2.

We also showed previously that GARLH is required for the synaptic clustering of the inhibitory scaffold gephyrin in the hippocampus (*Yamasaki et al., 2017*) and loss of gephyrin clustering was observed in γ2-null primary cortical neurons (*Essrich et al., 1998*). To determine if γ2 also plays a role in gephyrin clustering in the adult brain, we examined gephyrin distribution in γ2 GC-KO mice. In control mice, gephyrin clusters colocalized with the inhibitory presynaptic marker VGAT at the glomerular periphery (*Figure 4G and H*). By contrast, in the γ2 GC-KO mice, the fraction of gephyrin co-localized with VGAT was substantially reduced and the fraction of GluN1 signal co-localized with gephyrin was substantially reduced. (*Figure 4G and H*). Thus, γ2 directs the synaptic localization of gephyrin in the adult brain.

## δ inhibits synaptic localization of α6-containing GABA$_A$Rs

Two non-α/β subunits, γ2 and δ, are expressed in cerebellar granule cells, and γ2 is essential for GABA$_A$R assembly with GARLH/NL2 and synaptic localization in vivo. We next examined the role of δ in GABA$_A$R localization. δ assembles preferentially with α6-containing receptors (*Farrant and Nusser, 2005*). Interestingly, in δ KO cerebellar granule cells, an increase in the frequency and furosemide sensitivity of GABA$_A$R-mediated miniature IPSCs (mIPSCs) was reported (*Accardi et al., 2015*). Since furosemide selectively potentiates α6-containing GABA$_A$Rs, changes in furosemide sensitivity may suggest changes in α6 localization. To directly assess the role of δ in α6 localization, we analyzed the distribution of the α6 subunit in δ GC-KO cerebellum. We observed no obvious changes in the inhibitory presynaptic marker VGAT or the glomerular marker GluN1 in δ GC-KO cerebellum (*Figure 5A*). On the other hand, we observed weaker α6 signal in three δ GC-KO cerebella consistently by immunohistochemistry (*Figure 5A*). To confirm the specificity of the α6 signal, we obtained conventional α6 KO cerebellum (*Aller et al., 2003*), in which we observed an absence of α6 signal (*Figure 5A*). In addition, a reduction in expression of δ and β2/3 in total cerebellar lysate (*Figure 5—figure supplement 1A*) and δ signal by immunohistochemistry (*Figure 5—figure supplement 1B*) from α6 KO mice was confirmed, as published previously (*Jones et al., 1997*; *Nusser et al., 1999*). High-magnification images revealed that, in δ GC-KO mice, α6 formed clusters at the glomerular periphery that substantially overlapped with VGAT (*Figure 5B*). By contrast, in control littermates, α6 signal was diffuse and overlapped with GluN1 signal (*Figure 5B*). The fraction of α6 co-localized with VGAT was substantially increased in the granular layer of δ GC-KO mice, whereas the fraction of the entire glomerular marker GluN1 colocalized with α6 was reduced (*Figure 5C*). These results indicate that δ suppresses synaptic localization of α6 in the brain.

## δ suppresses an assembly pathway for α6-containing GARLHed GABA$_A$Rs

α6 localizes at synapses in δ GC-KO cerebellum (*Figure 5*), and γ2-containing GARLHed complexes are essential for synaptic GABA$_A$R activity (*Figure 4*). These results imply that, in the absence of δ, α6 incorporates with γ2 into GARLHed complexes.

To test this directly in vivo, we analyzed the compositions of GABA$_A$R complexes in δ GC-KO mice together with γ2 GC-KO and γ2/δ GC-DKO mice. Most strikingly, α6 incorporated into GARLHed complexes in cerebella from δ GC-KO mice, whereas α6 incorporated into GARLHless GABA$_A$Rs in cerebella from control, γ2 GC-KO and γ2/δ GC-DKO mice, on BN-PAGE (*Figure 6A*). The α6-containing GARLHed complex in δ GC-KO cerebella was eliminated in γ2/δ GC-DKO cerebella, supporting the earlier finding that γ2 is required for formation of the GARLHed complex (*Figure 6A*). Both cerebella from γ2 GC-KO and γ2/δ GC-DKO mice showed only a partial reduction in γ2 protein and the GARLHed complex, because the γ2 subunit and GARLHed complexes are also expressed in non-granule cell cerebellar neurons, including Purkinje cells (*Laurie et al., 1992*) (*Figure 4A*). In δ GC-KO cerebella, the amount of γ2 and NL2 in GARLHed complexes was increased relative to controls (*Figure 6A and B*). Consistent with this, in δ GC-KO cerebellum, all the essential

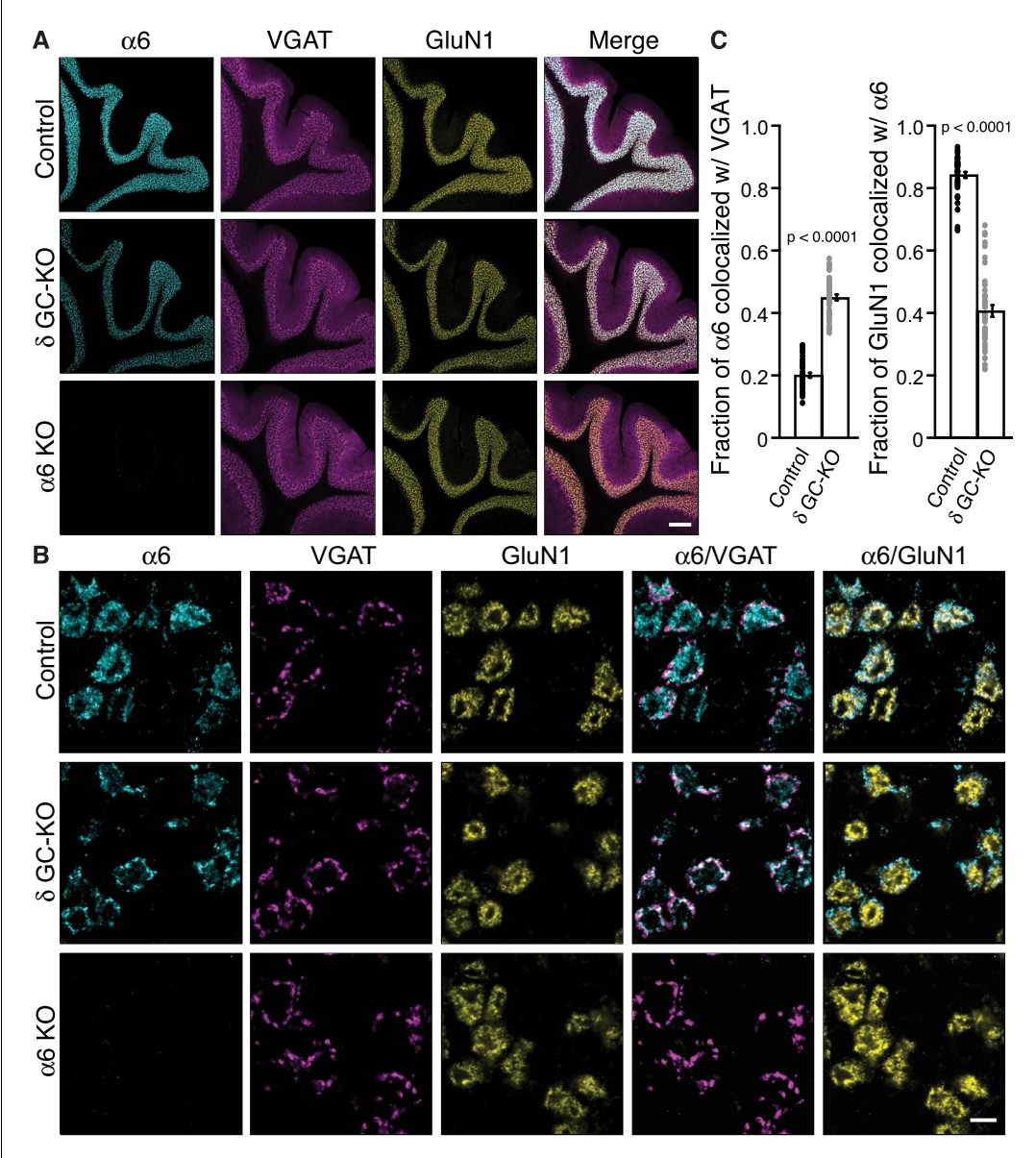

**Figure 5.** Delta inhibits synaptic localization of α6-containing GABA_ARs. (A–C) The distribution of α6 was examined in the cerebellum of δ GC-knockout (KO) and α6 KO mice. Inhibitory presynaptic VGAT and excitatory postsynaptic GluN1 were co-stained. (A) Low magnification images showed specific α6 signal in cerebellar granular layers in wild-type (Control) and δ KO mice, but not in α6 KO mice. The images are representative from three animals for each genotype. (B) High-magnification representative images showed VGAT around the glomeruli and GluN1 inside the glomeruli. In control mice, α6 signal was diffuse over the glomeruli, and overlapped substantially with GluN1. In contrast, in δ KO mice, α6 signal was largely confined to the peripheral glomeruli where it colocalized with VGAT. (C) The fraction of α6 signal co-localized with VGAT was increased in δ KO mice, whereas the fraction of GluN1 signal co-localized with α6 signal was reduced (n = 40–43 areas/3 animal each). Data are given as mean ± s.e.m.; p values were determined with student's t test. Scale bars: 200 μm (A), 5 μm (B).

The online version of this article includes the following figure supplement(s) for figure 5:

**Figure supplement 1.** α6 is required for expression of the δ subunit.

components of GARLHed complexes—GARLH4, NL2 and γ2—were upregulated without concomitant upregulation of α1, while α6 was only slightly decreased (*Figure 6C*). We also observed an increase in α1 and γ2 in GARLHless GABA_ARs on BN-PAGE in δ GC-KO cerebella, implying that in δ GC-KO cerebella, GARLH4 and/or NL2 becomes limiting for the GARLHed complex (*Figure 6A and*

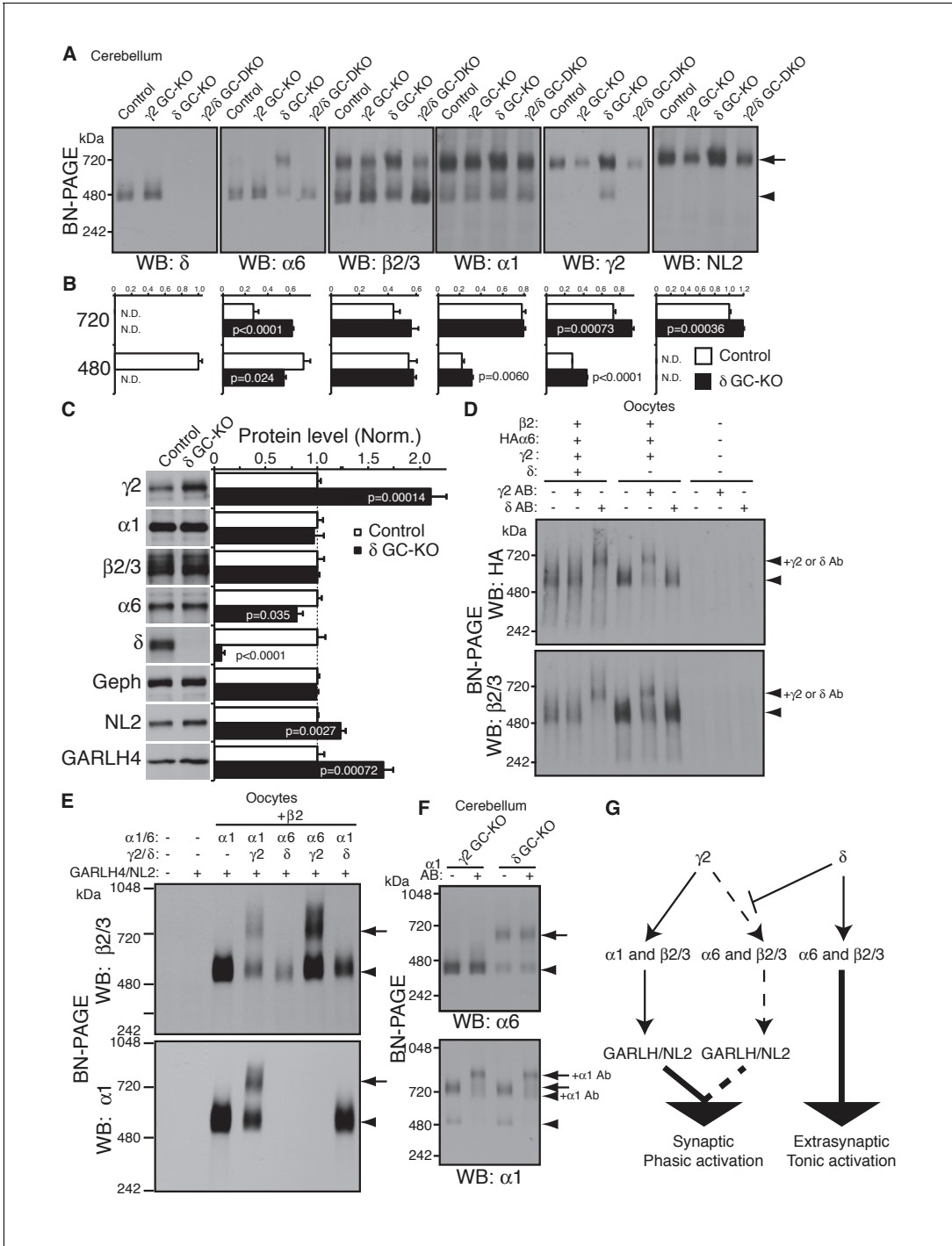

**Figure 6.** δ suppresses an assembly pathway for α6 with γ2, GARLH and NL2. (**A and B**) δ suppresses incorporation of α6 into GABA_AR/GARLH/NL2 complexes. (**A**) GABA_AR complexes in cerebella from mice of various genotypes were examined by BN-PAGE. In control, γ2 GC-KO and γ2/δ GC-double KO cerebella, α6 expressed predominantly at 480 kDa. In contrast, in δ KO cerebellum, α6 expressed predominantly at 720 kDa, and an increase in signals of β2/3, γ2 and NL2, but not α1, at 720 kDa was also observed. As expected, in γ2 GC-KO cerebellum, γ2 signal was reduced but not eliminated. The residual γ2 signal originates from other cell types in the cerebellum that still expressed γ2. The images are representative of three independent experiments. (**B**) Relative ratios of the 720 and 500 kDa complex in cerebella from control and δ GC-KO (n = 5 animals). Signal intensity of each band was measured. Relative ratios of bands at 720 and 480 kDa were calculated in control mice, and relative changes in each band intensity in δ GC-KO were estimated. Data are given as mean ± s.e.m.; p values were determined with student's t test. (**C**) Total protein expression in cerebella from δ KO mice. Results are shown relative to control littermates (n = 5 animals each). Elimination of δ expression was confirmed and α6 was modestly reduced. A substantial increase in γ2, GARLH4 and NL2 was observed without changes in other GABA_AR subunits (α1 and β2/3) or inhibitory synaptic

*Figure 6 continued on next page*

Figure 6 continued

marker protein Gephyrin (Geph). Data are given as mean ± s.e.m.; p values were determined with student's t test. (D) δ inhibits α6 assembly with γ2. GABA$_A$R complexes from cRNA-injected oocytes were examined by BN-PAGE. In oocytes expressing HAα6, β2 and γ2 with or without δ, HAα6 and β2/3 migrated at 520 kDa. Addition of anti-δ antibody, but not addition of anti-γ2 antibody, to membranes from HAα6/β2/γ2/δ-expressing oocytes shifted HAα6 and β2 signal upward, indicating preferential assembly of α6 with δ relative to γ2. On the other hand, when δ was not present, addition of anti-γ2 antibody shifted HAα6 and β2 signal upward. The images are representative of four independent experiments. (E) Membranes from oocytes injected with the indicated cRNAs (0.2 ng ea for α1, β2, γ2 and GARLH4; 0.5 ng for δ; 1.0 ng for α6 and NL2) were analyzed using BN-PAGE. Upon co-expression with γ2, but not δ, both α1 and α6 assembled with β2 and formed complexes with GARLH4 and NL2 at 720 kDa. The images are representative of two independent experiments. (F) α1 and α6 segregate into distinct complexes, even when both associate with GARLH/NL2. GABA$_A$R complexes in cerebella from γ2 GC-KO and δ GC-KO mice were examined by antibody shift assay on BN-PAGE. Addition of anti-α1 antibody shifted up α1 signal at 480 and 720 kDa in both genotypes. In contrast, in both genotypes, α6 signal was not shifted by addition of anti-α1 antibody. The images are representative of three independent experiments. The arrow and arrowhead indicate the GARLHed and GARLHless GABA$_A$R, respectively, and antibody-bound complexes are indicated. (G) δ suppresses an assembly pathway for α6-containing GARLHed GABA$_A$Rs. γ2 assembles with α1, β2/3 and GARLH/NL2 to mediate synaptic localization and phasic activation. Normally, δ sequesters α6, thereby suppressing γ2 interaction with α6. α6/δ-containing receptors do not interact with GARLH and neuroligin-2 (NL2), which are required for synaptic localization and phasic activation, and thus α6/δ-containing receptors localize at extrasynaptic sites and mediate tonic activation. In the absence of δ, α6 assembles with γ2, β2/3 and GARLH/NL2 to mediate synaptic localization and phasic activation.

*B*). Together, these results suggest that δ inhibits the incorporation of α6 into γ2-containing GARLHed complexes in vivo.

We next confirmed that δ is sufficient to inhibit α6 incorporation into γ2-containing GABA$_A$Rs in heterologous cells. To do this, we first used the antibody shift assay in cRNA-injected oocytes in the absence of GARLH/NL2. When HAα6 was expressed with β2, γ2 and δ, an anti-δ antibody, but not an anti-γ2 antibody, caused HAα6 signal to shift up (*Figure 6D*), indicating HAα6 oligomerization with δ but not γ2. In contrast, when HAα6 was expressed with β2 and γ2 without δ, the anti-γ2 antibody, but not anti-δ antibody, supershifted the HAα6 signal (*Figure 6D*). To confirm directly that α6 could incorporate into GARLHed complexes, we expressed combinations of α1, α6, γ2 and δ with both β2 and GARLH4/NL2 in oocytes and analyzed complexes by BN-PAGE. We found that GABA$_A$Rs associated with GARLH4/NL2, regardless of whether α1 or α6 was expressed, when coexpressed with γ2, but not δ (*Figure 6E*). These results indicate that δ is sufficient to inhibit the oligomerization of α6 with γ2.

Finally, we noted that although α1 and α6 preferentially segregate into distinct GABA$_A$Rs independent of non-α/β subunits (*Figure 2C*), in δ GC-KO cerebella, both α1 and α6 incorporate into GARLHed complexes (*Figure 6A*). To test if α1 and α6 segregate into distinct GARLHed complexes, we analyzed α1/α6 coassembly in γ2 GC-KO and δ GC-KO cerebella using the antibody shift assay. Addition of an anti-α1 antibody did not supershift α6 signal from GARLHless GABA$_A$Rs in γ2 GC-KO or from GARLHed complexes in δ GC-KO cerebella (*Figure 6F*). In contrast, addition of an anti-α1 antibody shifted up the α1 signal from GARLHed complexes in both cerebella (*Figure 6F*). This suggests that α1 and α6 remain largely segregated into separate complexes, even when both assemble with GARLH/NL2.

## Discussion

A long-standing question in the field of GABA$_A$R biology is the so-called 'combinatorial principle of receptor construction' (*Barnard et al., 1998*): What pentameric arrangements are favored in vivo, and what molecular rules determine these arrangements? This study reveals three novel rules governing the 'combinatorial principle' for native GABA$_A$R complexes. First, α1 and α6 subunits segregate into distinct GABA$_A$R pentamers independent of non-α/β subunits. Second, γ2 is required for GABA$_A$Rs to assemble with GARLH/NL2. Third, δ inhibits the incorporation of α6 into GARLHed complexes by sequestering it into GARLHless GABA$_A$Rs. These rules reveal the presence of an assembly pathway for α6-containing GARLHed complexes that is normally silenced by δ (*Figure 6G*). In the absence of δ, this pathway serves to increase inhibitory synaptic transmission (*Accardi et al., 2015*) by allowing α6-containing pentamers to assemble with GARLH/NL2 and localize to synapses (*Figure 6G*).

## Subunit compositions of distinct GABA_AR subtypes

In theory, a huge number of pentameric arrangements of GABA_AR subunits are possible. Our work reveals rules that help explain both why certain GABA_AR subtypes are favored, and why different subunits display distinct subcellular distributions. However, our results don't explain the atomic principles that must ultimately underlie these rules. For example, we found that intrinsic properties of α1 and α6 ensure their segregation into distinct pentamers independent of non-α/β subunits (*Figures 2C, D* and *6F*), but the atomic basis for this segregation was not investigated. The subunit arrangement in the prototypical α1/β2/γ2 pentamer is thought to be β2-α1-β2-α1-γ2 (*Baur et al., 2006*). In this case, it remains unclear how one α1 subunit could preferentially recruit another α1 subunit, given the intervening β2 subunit. One possibility is that non-adjacent α1 subunits actually make physical contacts, for example through their intracellular loops located between transmembrane domains 3 and 4. Another possibility is that the identity of each α subunit is conveyed allosterically via the intervening β2 subunit. For the α6/β2/δ pentamer, the situation is slightly different. One model for the subunit order for this pentamer is β2-α6-δ-α6-β2, with δ situated between both α6 subunits (*Baur et al., 2010*). In this case, it is possible that δ simply recruits both α6 subunits. However, this would still not explain why α6 subunits preferentially coassemble, excluding α1, even in the absence of δ (*Figure 2*). Thus far, a structural study of a β3 homopentamer lacking intracellular loops has provided critical information regarding the overall channel architecture, as well as atomic resolution descriptions of intersubunit β3-β3 contacts (*Miller and Aricescu, 2014*). High-resolution structures of GARLHless and GARLHed GABA_AR complexes with full-length proteins will ultimately be needed to gain atomic level insight into the assembly rules described here, and to identify domains responsible for α1 and α6 segregation.

We also found that α6 incorporates into γ2-containing GABA_ARs, which assemble with GARLH/NL2 (*Figure 6*) and localize at synapses (*Figure 5*) in δ GC-KO mice. Upregulation of γ2 in conventional δ KO mice was previously reported (*Tretter et al., 2001*), which we also observed in the δ GC-KO mice (*Figure 6C*). α1 and β3 were increased in conventional δ KO mice (*Tretter et al., 2001*) but were not changed in our cerebellar granule cell-specific δ KO mice (*Figure 6C*), perhaps because Cre expression under the *Gabra6* promoter is delayed until around P7. In δ GC-KO mice, we also observed an increase in the other essential components of the cerebellar GARLHed GABA_AR complex, namely GARLH4 and NL2 (*Figure 6C*). One possibility is that, in the absence of δ, the synthesis of γ2, GARLH4 and NL2 is increased to accommodate α6 that is no longer sequestered by δ. Alternatively, without δ, excess α6 might bind to and stabilize γ2, GARLH4 and/or NL2, thus increasing protein levels independent of changes in synthesis. Further studies will be needed to address these details.

## Synaptic targeting of α6-containing GABA_ARs

In δ GC-KO mice, α6 associates with γ2, GARLH4 and NL2 and is redistributed to synapses, strongly suggesting that α6 synaptic localization, like α1 synaptic localization, requires γ2 and GARLH4. To test this formally, future studies should assess the ability of α6 to localize to synapses in the absence of γ2 and/or GARLH4 in δ KO mice. Similar to α6, α4 is proposed to localize to extrasynaptic sites and also assembles with δ in hippocampus (*Jechlinger et al., 1998*; *Jones et al., 1997*; *Wongsamitkul et al., 2016*). It would be interesting to examine whether, in δ KO mice, α4 also incorporates into GARLHed complexes and is targeted to inhibitory synapses.

Our findings are also consistent with the reported increase in mIPSC sensitivity to furosemide, which preferentially inhibits α6-containing GABA_ARs, in δ KO mice (*Accardi et al., 2015*). Accardi and colleagues also reported an increase in mIPSC frequency, but not amplitude, in δ KO mice. While increases in mIPSC frequency are sometimes attributed to presynaptic alterations, the authors posited a second possibility, namely that the number of inhibitory synapses is increased in the absence of δ. Given our finding that α1 and α6 segregate into distinct pentamers even in the absence of δ, one possibility is that in δ KO mice, α6-containing pentamers actually localize to and activate a distinct set of inhibitory synapses. Further studies will be needed to address this possibility.

## Can manipulation of an extrasynaptic subunit modulate synaptic strength?

The major GABA$_A$R subtypes in the brain accommodate only one non-α/β subunit, and thus incorporation of δ into a pentamer precludes incorporation of γ2 and blocks assembly with GARLH/NL2. This suggests the intriguing hypothesis that changes in δ expression—for example, by ethanol or seizure activity (*Cagetti et al., 2003*; *Liang et al., 2006*; *Marutha Ravindran et al., 2007*; *Peng et al., 2004*; *Zhang et al., 2007*)—could control the ratio of GARLHed complexes and GARLHless pentamers in vivo, and thus alter inhibitory synaptic strength. Supporting this, a marked increase in α6-containing GABA$_A$R-mediated IPSCs in cerebellar granule cells was observed in δ KO mice (*Accardi et al., 2015*). Were this hypothesis fully substantiated, it would provide an opportunity to pharmacologically control inhibitory transmission by targeting the extrasynaptic δ subunit. Future studies are required to examine δ expression as a potential drug target.

## Materials and methods

### Antibodies

| Protein | RRID | Species | Provider | Cat# | Epitope (AA) | Epitope (domain) |
|---|---|---|---|---|---|---|
| GABARβ2/3 | AB_309747 | Mouse | Millipore | 05–474 | Not specified | Not specified |
| GABARα1 | AB_2108811 | Mouse | Neuromab | 75–136 | AA355-394 | Cytoplasmic loop (intracellular) |
| PSD95 | AB_2307331 | Mouse | Neuromab | 75–028 | AA77-299 | PDZ1 and 2 |
| HA | AB_2314672 | Mouse | Covance | MMS-101P | | HA peptide |
| GABARα1 | AB_310272 | Rabbit | Millipore | 06–868 | AA1-15 (mature protein) | NTD (extracellular) |
| GABARα6 | AB_11212626 | Rabbit | Millipore | AB5453 | | Cytoplasmic loop (intracellular) |
| GABARα6 | AB_2039868 | Rabbit | Alomone | AGA-004 | AA20-37 | Extracellular |
| GABARγ2 | AB_11211236 | Rabbit | Millipore | AB5559 | | Cytoplasmic loop (intracellular) |
| GABARδ | AB_672966 | Rabbit | Millipore | AB9752 | | NTD (extracellular) |
| HA | AB_390918 | Rat | Roche | 11 867 431 001 | | HA peptide |
| GARLH4 | N.A. | Rabbit | Yamasaki et al | N.A. | AA195-247 | CTD (Intracellular) |
| NL2 | AB_993011 | Rabbit | Synaptic Systems | 129 202 | AA732-749, AA750-767 | CTD (Intracellular) |
| VGAT | AB_887873 | Guinea pig | Synaptic Systems | 131 004 | AA2-155 | N-terminus |
| GluN1 | AB_396353 | Mouse | BD Pharmingen | 556308 | AA660-811 | Extracellular |
| Gephyrin | AB_2232546 | Mouse | Synaptic systems | 147 021 | | N-terminus |
| Gephyrin | AB_397930 | Mouse | BD Pharmingen | 610585 | AA569-726 | C-terminus |

### Plasmids

GARLH4, NL2 and GABA$_A$R subunit α1, α6, β2, γ2 and δ cDNAs (Open Biosystems) were cloned into appropriate vectors (pGEM-HE or gateway entry vectors (Invitrogen)). Epitope tags were inserted using Quick Change mutagenesis (Stratagene, La Jolla, CA). The concatenated constructs were modifications of constructs reported previously (*Baur et al., 2006*) and were generated using MultiSite Gateway Technology (Invitrogen).

### Animals

All animal handling was in accordance with a protocol (#11029) approved by the Institutional Animal Care and Use Committee (IACUC) of Yale University. Animal care and housing was provided by the Yale Animal Resource Center (YARC), in compliance with the Guide for the Care and Use of Laboratory Animals (National Academy Press, Washington, D.C., 1996). Wild-type (C57BL/6J, Stock# 000664, RRID:IMSR_JAX:000664), the conditional *Gabrd* (Stock # 023836, RRID:IMSR_JAX:023836), the conditional *Gabrg2* (Stock# 016830), and the transgenic CreERT mouse under the CAG promoter (Stock# 004682, RRID:IMSR_JAX:004682) were obtained from the Jackson Laboratory. The

transgenic Cre mouse under the *Gabra6* promoter (ID# 015966-UCD, RRID:MMRRC_015966-UCD) and the *Gabra6* knockout (ID# 015968-UCD, RRID:MMRRC_015968-UCD) were obtained from MMRRC. Oocytes were harvested from *Xenopus laevis* (Product number: LM00535MX) obtained from Nasco.

## Electrophysiology and surface expression using *Xenopus laevis* oocytes

Two-electrode voltage clamp (TEVC) recordings and measurements of surface expression were performed as described (*Tomita et al., 2005*; *Tomita et al., 2004*; *Zerangue et al., 1999*; *Zhang et al., 2009*). Briefly, cDNAs were subcloned into pGEM-HE vector and cRNA was transcribed in vitro using T7 mMessage mMachine (Ambion). TEVC analysis was performed 3–5 days after injection at room temperature in ND96 containing (in mM): 90 NaCl, 2 KCl, 1.8 CaCl2, 1 MgCl2, 5 HEPES (pH 7.5). The membrane potential was held at −40 mV. HA-tagged proteins at the cell surface were labeled with Rat anti-HA antibody (Roche) and horseradish-peroxidase (HRP) conjugated secondary antibody (GE Health), and measured with a chemiluminescence assay.

## Blue native-PAGE and antibody shift

BN-PAGE was performed as described previously (*Kim et al., 2010*; *Schägger et al., 1994*). Briefly, membrane fractions from cRNA-injected oocytes or the mouse cerebellum were solubilized with 0.5% Triton X-100 or 1% Lauryl Maltose-neopentyl glycol, respectively. For the antibody shift assay, the samples were incubated with the indicated antibody for 2 hr. The solubilized proteins were then resolved on SDS-PAGE or BN-PAGE (4–12%), which was followed by western blot analysis. Molecular weights on BN-PAGE were determined using the NativeMark Unstained Protein Standard (Life Technologies). The gels were scanned using a scanner (EPSON PERFECTION 4490 PHOTO) at a resolution of 600 dpi. Scanned images were cropped and assembled on Illustrator (Adobe) for printing without any further adjustment. For quantification, each gel was run with a series of diluted samples to generate a standard curve for each protein detected by western blotting, and signal intensity of each band was measured using ImageJ (NIH) and quantified with the standard curve.

## Cerebellar granule cell culture

Primary cultured cerebellar granule cells were prepared as described (*Zhang et al., 2009*). Briefly, P7 mice were anesthetized on ice and decapitated. Cerebella were dissected, treated with trypsin, and cells were plated on poly-D-lysin (PDL) treated glass coverslips at a density of ~$1 \times 10^6$ cells/cm$^2$ and grown in a humidified incubator at 37°C, 5% $CO_2$. Neurons were treated with 4-hydroxytamoxifen from DIV1.5 to DIV3 (400 nM, Sigma).

## Immunohistochemistry

Adult mice were deeply anesthetized with pentobarbitol (100 mg/kg) and perfused transcardially with 4% paraformaldehyde in 0.1 M phosphate buffer pH 7.4. After post-fixation, 30–40 µm sections were prepared using a vibratome (Leica). Sections were incubated with 1 mg/ml pepsin (DAKO) in 0.2 N HCl for 3–10 min at 37°C and washed with PBS, stained with appropriate antibodies and imaged by confocal microscopy (Zeiss 710) (*Straub et al., 2011*). Image quantification was performed using ImageJ.

Quantification of co-localization was performed using Mander's coefficient analysis through the JACoP plugin in ImageJ (*Bolte and Cordelières, 2006*).

## Cerebellar slice synaptic electrophysiology

Mice (P25-P35) were deeply anesthetized with isoflurane and euthanized by decapitation. Brains were rapidly extracted and transferred to ice cold artificial cerebrospinal fluid (ACSF, containing (in mM):120 NaCl, 2 KCl, 2 CaCl2, 1.2 MgSO4, 1.2 KH2PO4, 26 NaHCO3, and 11 glucose; equilibrated with 95% O2, 5% CO2). Sagittal cerebellar sections (200 µm) were prepared using a vibratome (Leica). Granule cells were identified visually using an upright microscope (Olympus), and recordings were performed in oxygenated ACSF at room temperature. Patch pipets had a resistance of 5–10 MΩ and were filled with an internal solution containing the following (in mM): 81 CsSO4, 4 NaCl, 2 MgSO4, 0.02 CaCl2, 0.1 BAPTA, 15 HEPES, 15 Dextrose, 3 Mg-ATP, 0.1 Na-GTP (pH 7.2, adjusted with CsOH). To isolate GABA$_A$R mediated spontaneous inhibitory postsynaptic currents (sIPSCs),

AP-5 (100 µM) and CNQX (20 µM) were added to the external solution. sIPSCs were recorded from cerebellar granule cells in whole-cell configuration, using a Multiclamp 700B amplifier (Axon Instruments), at a holding potential of $-10$ mV. In these conditions, sIPSCs manifested as outward current. To confirm that sIPSCs were GABA$_A$R-mediated, 100 µM picrotoxin was applied to the external solution after each recording. Online data acquisitions were performed using the Clampex program (Axon Instruments). Signals were filtered at 2 kHz and digitized at 25 kHz. Offline analysis was performed using IgorPro (WaveMetrics, Inc, Lake Oswego, OR, USA) and Mini Analysis (http://www.syn-aptosoft.com, Decatur, GA, USA). For quantification of amplitude and average traces, individual events were averaged. For quantification of frequency, events from two to four neurons were divided into bins (n = 4), and average values from each bin were measured. Reported values are the average of averages from each bin. All chemicals were obtained from Tocris Cookson or Sigma.

### Statistical analysis

Quantification and statistical details of experiments can be found in the figure legends or Method Details section. All data are given as mean ± s.e.m. Statistical significance between means was calculated using Student's t test. The number of independent experiments is indicated in each figure legend.

## Acknowledgements

The authors thank Pietro De Camilli, Angus Nairn, Peter Aronson, Michael Higley, Houqing Yu, Ania Puszynska and members of the Tomita lab for helpful discussions. We thank Dr. Erwin Sigel for original GABA$_A$R concatenated constructs, Dr. Janet L Fisher for GABA$_A$R α6 cDNA, Dr. Louis Reichardt for sharing transgenic Cre mice under the *Gabra6* promoter through the MMRRC, Dr. Bernhard Luscher and Dr. Jamie Maguire for conditional γ2 mice and δ mice, respectively, through the Jackson laboratory. The monoclonal antibodies were obtained from the University of California Davis/National Institutes of Health NeuroMab Facility (NIH U24NS050606). ST is supported by NIH MH077939, MH104984 and Yale University, JSM is supported by NIH F30 MH099742 and the NIH T32GM007205, TY is supported by the Uehara Memorial Foundation, and NHC is supported by NIH F30 MH113299, CTSA TL1TR000141 and NIH/NIGMS T32 GM007205.

## Additional information

### Funding

| Funder | Grant reference number | Author |
|---|---|---|
| NIH Clinical Center | F30 MH099742 | James S Martenson |
| NIH Clinical Center | GM007205 | James S Martenson Nashid H Chaudhury |
| NIH Clinical Center | T32GM007205 | James S Martenson |
| NIH Clinical Center | T32 GM007205 | Tokiwa Yamasaki |
| NIH Clinical Center | TL1TR000141 | Nashid H Chaudhury |
| NIH Clinical Center | F30MH113299 | Nashid H Chaudhury |
| NIH Clinical Center | MH077939 | Susumu Tomita |
| NIH Office of the Director | MH104984 | Susumu Tomita |

The funders had no role in study design, data collection and interpretation, or the decision to submit the work for publication.

### Author contributions

James S Martenson, Data curation, Formal analysis, Validation, Visualization, Methodology, Writing—review and editing; Tokiwa Yamasaki, Nashid H Chaudhury, Data curation, Formal analysis, Validation, Visualization, Methodology; David Albrecht, Data curation, Formal analysis, Validation,

Methodology; Susumu Tomita, Conceptualization, Resources, Supervision, Funding acquisition, Methodology, Writing—original draft, Writing—review and editing

## Author ORCIDs

Susumu Tomita (iD) http://orcid.org/0000-0001-8344-259X

## Ethics

Animal experimentation: All animal handling was in accordance with a protocol (#11029) approved by the Institutional Animal Care and Use Committee (IACUC) of Yale University. Animal care and housing was provided by the Yale Animal Resource Center (YARC), in compliance with the Guide for the Care and Use of Laboratory Animals (National Academy Press, Washington, D.C., 1996).

## Decision letter and Author response

Decision letter https://doi.org/10.7554/eLife.27443.sa1
Author response https://doi.org/10.7554/eLife.27443.sa2

## Additional files

### Supplementary files

• Transparent reporting form

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
