## [Decision Letter]

Thank you for submitting your article "Assembly rules for GABA_A_ receptor complexes in the brain" for consideration by *eLife*. Your article has been reviewed by three peer reviewers, one of whom is a member of our Board of Reviewing Editors and the evaluation has been overseen by the Reviewing Editor and Eve Marder as the Senior Editor. The following individual involved in review of your submission has agreed to reveal her identity: Mary B Kennedy (Reviewer #1).

The reviewers have discussed the reviews with one another and the Reviewing Editor has drafted this decision to help you prepare a revised submission.

Summary:

The subunit composition of GABA receptors dictates the subcellular localization and properties of the receptors. Only a few of the theoretically possible subunit combinations exist in the brain, but it is largely unresolved how the selection of the receptor components is regulated. The manuscript by Martenson et al., tackles this important question. The authors analyzed both recombinant and native GABA_A_ receptor complexes by blue native polyacrylamide gel electrophoresis and antibody-shift assays to unravel which subunits can assemble into a receptor, which subunits exclude each other and which subunits are critical for forming a synaptic complex. They substantiated their findings in cerebellar granule cells genetically manipulated to lack specific receptor subunits. Based on the results, the authors put forward three "assembly rules" for subunits of GABA-A receptors into pentamers with associated auxiliary subunits, that help to explain segregation of subunits between synaptic (phasic inhibition) and extrasynaptic (tonic inhibition) localizations. The conclusions represent a significant and important advance in understanding the assembly of inhibitory receptor complexes. The experiments are logically designed and support the conclusions. However, there are deficits in the analysis of the data and its presentation that must be corrected before publication in *eLife*.

Essential revisions:

1) The information provided about methods, replication, and statistics in the figure legends and/or methods section is inadequate in many respects. Each figure legend should list the number of times each experiment was repeated, and a measure of the agreement between repetitions. This is particularly lacking in the methods and legends for figures showing results of BN-PAGE. The methods provide only the general statement that all experiments were replicated. The number of replications should be listed for each part of each experimental figure. For example, the methods do not state, nor do earlier cited papers, how the gels were scanned and how the images were processed. What was the make and model of the scanner? What software was used to adjust the images for printing and to quantify the images?

2) In Figure 2, an asterisk marks a "non-specific band". The figure should state the source of the non-specific band. Was it present in a blot of secondary alone? How do the authors know that it is "non-specific?"

3) It is well established that γ2 is essential for the synaptic localization of GABA_A_ receptors. Three of the authors of the current manuscript contributed to a 2017 publication identifying GARLHs as GABA_A_ receptor components that link the core GABA_A_ receptor to NL2 at inhibitory synapses (Yamasaki et al., 2017). Within the earlier study, it was shown that γ2 was required to bridge GABA_A_ receptors and GARLHs/NL2 upon heterologous expression. In the current work the authors refer to this finding and show that most of the GARLHed complexes in the cerebellum contain γ2 and that a deletion of γ2 in cerebellar granule cells in culture eliminated incorporation of α1 and β2/3 into GARLHed complexes. Furthermore, using γ2 GC-KO mice they provide evidence that γ2 is essential for the synaptic localization of GABA_A_ receptors in cerebellar granule cells. However, the γ2 GC-KO mice have been used in the previous study too, and a strongly reduced expression of LH4 and NL2 in cerebellar extracts and of LH4, NL2 and α1 in glomerular postsynapse-enriched fractions has been reported (Yamasaki et al., 2017). The current manuscript should refer to these results as well, e.g. in the sections describing the experiments presented in Figures 3 and 4. The expression levels of LH4 and NL2 in cerebellar granule cell cultures with and without γ2 (Figure 3C) need to be determined and shown. The authors should point out that their conclusion that the association of native GABA_A_ receptors with GARLHs requires γ2 is based on a combination of findings described in the current work and the previous publication (Yamasaki et al., 2017).

4) The authors show that intrinsic signals in α1 and α6 prevent their incorporation into the same receptor. The authors discuss that it is unclear how these intrinsic signals function since α1 and α6 are separated by β and δ, respectively, within the pentamer. The conclusion that intrinsic properties prevent the two subunits from assembling together is justified. However, to get more insight into the intrinsic signals it would be useful to know if the authors have tried to narrow down regions within α1 and α6 that are involved in the segregation of the subunits.

The authors provide evidence that γ2 can assemble into α6-containing receptors in the absence of δ and connect them with synapses suggesting that the strength of inhibitory synapses could be regulated by the level of δ expression. It would also be interesting to include data as to whether the effect of δ is specific for α6-containing receptors or whether δ is able to assemble into α1-containing receptors in the absence of γ2. These two sets of experiments will not be required for publication, but the paper would be more interesting if the authors can include such information.

5) Figure 4B and 4F are not quantified. From the images shown in 4B, I had a hard time seeing the changes of alpha1 localization in the gamma2 KO. The Ephys phenotype of the KO is very strong as shown in Figure 4C. The synaptic signals in Figure 4B and 4F should be quantified. The same is true for the glomerular synaptic images in Figure 5.

---

## [Author Response]

Essential revisions:1) The information provided about methods, replication, and statistics in the figure legends and/or methods section is inadequate in many respects. Each figure legend should list the number of times each experiment was repeated, and a measure of the agreement between repetitions. This is particularly lacking in the methods and legends for figures showing results of BN-PAGE. The methods provide only the general statement that all experiments were replicated. The number of replications should be listed for each part of each experimental figure. For example, the methods do not state, nor do earlier cited papers, how the gels were scanned and how the images were processed. What was the make and model of the scanner? What software was used to adjust the images for printing and to quantify the images?

We apologize that information about methods, replication, and statistics was inadequate. We now provided all the information in detail as below.

We added the number of times each experiment was independently repeated. Furthermore, we emphasized that each figure is representative of the independent replicates, in order to indicate a high measure of agreement between repetitions in each figure legend.

We now state, “The gels were scanned using a scanner (EPSON PERFECTION 4490 PHOTO) at a resolution of 600 dpi. Scanned images were cropped and assembled on Illustrator (Adobe) for printing without any further adjustment. For quantification, each gel was run with a series of diluted samples to generate a standard curve for each protein detected by western blotting, and signal intensity of each band was measured using ImageJ (NIH) and quantified with the standard curve.”

2) In Figure 2, an asterisk marks a "non-specific band". The figure should state the source of the non-specific band. Was it present in a blot of secondary alone? How do the authors know that it is "non-specific?"

We apologize for not describing this. We consistently observe this band around 100 kDa when analyzing oocyte extracts using SDS-PAGE and western blotting, regardless of the primary antibody used. Thus, the asterisk-marked band reflects an endogenous oocyte protein, and not a heterologously expressed GABA_A_R subunit. And we called the band a “non-specific band”. We clarified this point.

3) It is well established that γ2 is essential for the synaptic localization of GABA_A_ receptors. Three of the authors of the current manuscript contributed to a 2017 publication identifying GARLHs as GABA_A_ receptor components that link the core GABA_A_ receptor to NL2 at inhibitory synapses (Yamasaki et al., 2017). Within the earlier study, it was shown that γ2 was required to bridge GABA_A_ receptors and GARLHs/NL2 upon heterologous expression. In the current work the authors refer to this finding and show that most of the GARLHed complexes in the cerebellum contain γ2 and that a deletion of γ2 in cerebellar granule cells in culture eliminated incorporation of α1 and β2/3 into GARLHed complexes. Furthermore, using γ2 GC-KO mice they provide evidence that γ2 is essential for the synaptic localization of GABA_A_ receptors in cerebellar granule cells. However, the γ2 GC-KO mice have been used in the previous study too, and a strongly reduced expression of LH4 and NL2 in cerebellar extracts and of LH4, NL2 and α1 in glomerular postsynapse-enriched fractions has been reported (Yamasaki et al., 2017).

We wish to thank the reviewers for their careful reading of our previous manuscript (Yamasaki et al., 2017).

The current manuscript should refer to these results as well, e.g. in the sections describing the experiments presented in Figures 3 and 4.

We now refer to the previous results in Yamasaki et al., 2017 in Figures 3 and 4.

The expression levels of LH4 and NL2 in cerebellar granule cell cultures with and without γ2 (Figure 3C) need to be determined and shown.

We are willing to do this experiment, which requires P7 pups homozygous for the conditional γ2 allele and carrying a CreERT transgene. Unfortunately, despite multiple attempts at breeding, we have not yet obtained pups of the required genotype, and thus we will not be able to provide this result within the two-month window for revision. We can perform this experiment immediately as soon as the pups become available.

That said, we feel that the information this experiment would provide is not necessary to draw our current conclusion. The novelty of this figure is the loss of the 720 kDa complex in cerebellar granule cell cultures lacking γ2. This is the first demonstration that γ2 is required for the formation of the 720 kDa complex in vivo. Because the expression levels of LH4 and NL2 in cerebella from adult γ2 knockout mice were already evaluated in Yamasaki et al., 2017, we feel our conclusions are fully supported without the additional experiments requested. Thank you for considering our situation.

The authors should point out that their conclusion that the association of native GABA_A_ receptors with GARLHs requires γ2 is based on a combination of findings described in the current work and the previous publication (Yamasaki et al., 2017).

We now point this out clearly by stating, “Combining our new finding that γ2 is required in vivo for assembling the native GABA_A_R complex (Figure 3C) with the finding from Yamasaki et al., 2017 that γ2 is required for reconstituting the native GABA_A_R complex in a heterologous system, we conclude that the association of native GABA_A_Rs with GARLHs requires γ2.”

4) The authors show that intrinsic signals in α1 and α6 prevent their incorporation into the same receptor. The authors discuss that it is unclear how these intrinsic signals function since α1 and α6 are separated by β and δ, respectively, within the pentamer. The conclusion that intrinsic properties prevent the two subunits from assembling together is justified. However, to get more insight into the intrinsic signals it would be useful to know if the authors have tried to narrow down regions within α1 and α6 that are involved in the segregation of the subunits.

We have not tried to identify the intrinsic signal domains in α1 and α6. Mapping the regions within α1 and α6 that are involved in the segregation of the subunits using chimeric proteins would provide valuable information. However, interpretations of results using chimeric proteins would be difficult without a high-resolution atomic structure of this complex, because mutations might induce global changes in structure that may cause disruption of the intrinsic signal indirectly. Thus, we left a detailed dissection of the segregation of α1 and α6 for future studies.

The authors provide evidence that γ2 can assemble into α6-containing receptors in the absence of δ and connect them with synapses suggesting that the strength of inhibitory synapses could be regulated by the level of δ expression. It would also be interesting to include data as to whether the effect of δ is specific for α6-containing receptors or whether δ is able to assemble into α1-containing receptors in the absence of γ2. These two sets of experiments will not be required for publication, but the paper would be more interesting if the authors can include such information.

Thank you for this excellent suggestion. Thank you also for recognizing that, although the experiment would provide valuable information, it is not essential for our main argument. To assess the ability of α1 and δ to interact in the absence of γ2, the best experiment would be an antibody shift analysis using anti-α1 and anti-δ antibodies for shift and western blotting, respectively. Unfortunately, the combination of anti-α1 and anti-δ antibodies showed significant background, a common issue in antibody shift assays on BN-PAGE. Because the result was inconclusive, we did not include it.

5) Figure 4B and 4F are not quantified. From the images shown in 4B, I had a hard time seeing the changes of alpha1 localization in the gamma2 KO. The Ephys phenotype of the KO is very strong as shown in Figure 4C. The synaptic signals in Figure 4B and 4F should be quantified.

Thank you for this feedback. We have revised the figure according to your comments, and believe the new figure is a substantial improvement over the original. To quantify the images in Figure 4 in the same way used for Figure 5, we performed similar experiments and quantified staining usingγ2 KO and Cre-negative control littermates. This result showed a reduction in the co-localization of α1 (Figure 4B) and gephyrin (Figure 4F) with VGAT in γ2 GC-KO, suggesting a reduction in synaptic α1 and gephyrin in γ2 KO. In addition, α1 colocalized more frequently with the glomeruli marker GluN1 in the cerebellar granular layer of γ2 GC-KO, whereas gephyrin co-localization with GluN1 was reduced. This result suggests that, in γ2 GC-KO, α1 redistributes diffusely on the cerebellar glomeruli, whereas gephyrin partly redistributes outside the glomeruli.

The same is true for the glomerular synaptic images in Figure 5.Supplemented comment (provided after an author query): "In Figure 5, the intensity of a6 signal is reduced in the delta KO. Is that a real effect? Does this affect the colocalization analysis? From these images, it appears that the amount of a6 receptors apposing the VGAT positive presynaptic terminals is similar if not reduced in the delta KO. If so, how can we explain the increase in IPSC frequency in the KO?"

The concern is that the absolute quantity of alpha6 might be reduced. More explanation and/or absolute quantification of alpha6 staining would help.

In Figure 5, the intensity of a6 signal is reduced in the delta KO. Is that a real effect?

Thank you for pointing this out, and sorry that we did not clarify in detail. It is difficult to meaningfully quantify absolute signal intensity between sections because a standard curve is unavailable for immunohistochemistry.

Nonetheless, we saw the weaker α6 signal consistently in three different δ GC-KO animals by immunohistochemistry (Figure 5A is representative), and measured a significant decrease in α6 levels in δ GCKO animals by SDS-PAGE and western blotting (Figure 6C), so we believe the effect is real.

Does this affect the colocalization analysis?

No. The reduction in α6 signal does not affect our colocalization analysis using the Mander’s colocalization coefficient (Dunn, Kamocka, and McDonald, 2011; Bolte and Cordelieres, 2006). Our conclusion from Figure 6 is that a greater portion of α6 is concentrated at synapses in δ GC-KO relative to controls. This conclusion is independent of the absolute expression level of α6. Perhaps, the label on Y-axis of this figure in the original manuscript was not clear. We revised the Y-axis label as “Fraction of α6 colocalized with VGAT.”

From these images, it appears that the amount of a6 receptors apposing the VGAT positive presynaptic terminals is similar if not reduced in the delta KO. If so, how can we explain the increase in IPSC frequency in the KO?

As stated above, it is difficult to meaningfully quantify absolute signal intensity between sections because a standard curve is unavailable for immunohistochemistry. Thus, we are not able to compare the absolute signal intensity of α6 colocalized with VGAT between control and δ GC-KO mice in a confident manner. Instead, we used the Mander’s colocalization coefficient (Dunn, Kamocka, and McDonald, 2011; Bolte and Cordelieres, 2006) to assess the fraction of total α6 signal colocalized with VGAT. Importantly, no method is available to derive IPSC frequency from synaptic clustering detected by immunohistochemistry. The point of this figure is that α6 localizes to synapses more than extrasynaptic sites in δ GC-KO mice. This finding is totally novel.